# The host–guest inclusion driven by host-stabilized charge transfer for construction of sequentially red-shifted mechanochromic system

Dongdong Sun [1], Yong Wu[1], Xie Han[1,2] & Simin Liu [1,2] ✉

Developing more extensive methods to understand the underlying structure-property relationship of mechanochromic luminescent molecules is demanding but remains challenging. Herein, the effect of host-guest interaction on the mechanochromic properties of organic molecules is illustrated. A series of pyridinium-functionalized triphenylamine derivatives show bathochromic-shifted emission upon mechanical stimulation. These derivatives bind to cucurbit[8]uril to form homoternary host-guest inclusion complexes through host-stabilized intermolecular charge transfer interactions. Remarkably, the homoternary complexes exhibit longer emission than that of free guests in the solid state (even longer than ground guests), and a further bathochromic-shifted emission is observed upon grinding. Additionally, a heteroternary complex constructed through the encapsulation of pyrene (donor) and pyridinium (acceptor) guest pair in cucurbit[8]uril also displays the mechanochromic luminescent property. This work not only discloses the effect of host-guest inclusion on the mechanochromic property of organic molecules, but also provides a principle and a facile way to design the sequentially red-shifted mechanochromic materials.

Mechanochromic luminescent (MCL) materials, referring that the photophysical properties including emission wavelength, emission lifetime, and quantum yield are sensitive to external mechanical stimulation[1]. As one of the most promising smart materials, MCL materials have attracted great attention and been widely used in construction of sensors and information storage devices[2]. More and more organic MCL molecules have been developed such as distyrylanthracene[3], oligo(p-phenylene vinylene)[4,5], boron complexes[6], triphenylamine (TPA), tetraphenylethylene derivatives[7,8], and others[9,10], along with the proposed design strategies including cation–anion interaction[11,12], donor–acceptor type compounds[13,14], doping[15,16], site effect of substituent[17,18], etc. These approaches mainly focused on the construction of different

mechanoresponsive systems through molecular structure modification. However, even with the same backbone, the subtle change in molecular structure could make great difference on the photophysical properties and mechanoresponsive behavior, highlighting the need to understand the structure-property relationship of specific design methods[19,20]. In addition, most of the MCL systems reported thus far were generated by transitioning from a crystalline state to an amorphous or a new crystalline form[21,22]. However, the strict conditions required for crystal cultivation limit the applicability of MCL materials. Therefore, developing amorphous MCL materials is of great significance.

Numerous pioneering works have shown that both molecular packing and conformation are crucial for the mechanoresponsive

[1]School of Chemistry and Chemical Engineering, Wuhan University of Science and Technology, Wuhan 430081, PR China. [2]The State Key Laboratory of Refractories and Metallurgy, Institute of Advanced Materials and Nanotechnology, Wuhan University of Science and Technology, Wuhan 430081, PR China. ✉e-mail: liusimin@wust.edu.cn

behavior of materials in the solid state[23,24]. From this perspective, besides ingenious molecular designs, other strategies that could alter the packing arrangement and/or conformation in the solid state may also be suitable for constructing MCL materials. The host–guest interaction has been proved to be a powerful tool for regulating the packing arrangement and conformation of guests. On one hand, encapsulating chromophores on guest molecules by suitable macrocyclic hosts enables the conversion of monomer, dimer, and supramolecular polymer/assembly, while tuning the packing arrangement usually leads to intriguing emission property[25–30]. On the other hand, host-induced folded or planarized conformation of guests also results in appealing photophysical property[31–35]. Additionally, the host–guest complex with high binding affinity could maintain the specific binding mode in the solid state. Even in an amorphous state, when part of the twisted guest is located outsides the confined cavity, the host–guest complex may still show responsiveness toward mechanical stimulation. Overall, the host–guest complexation may be an appropriate candidate for designing MCL materials.

Cucurbit[$n$]uril (CB[$n$], $n$ = 5–8, 10) is composed of $n$ glycoluril units bridged by $2n$ methylene and possesses a symmetric structure with a confined cavity, showing high binding affinity towards cationic guests[36–38]. The past few years have witnessed the great achievements of CB-based luminescent materials, including tunable luminescent materials and organic room temperature phosphorescence materials[39–41]. In addition, the supramolecular structure of donor–acceptor (D-A) systems united through intermolecular charge transfer (CT) interaction has attracted tremendous attention and provided a prospective way to induce unique luminescent systems[42–44]. Noncovalent D-A structures usually exhibit specific charge transfer emission that could be modulated by changing the donor or acceptor groups[45,46]. Besides, this supramolecular method has been utilized to construct striking MCL materials, in which the mechanical stimulation could modulate the packing of the assembled D-A groups[15,16,47]. CB[$n$] could dramatically enhance the intermolecular CT interaction between donors/acceptors to form the heteroternary inclusion complexes[48]. However, the mechanoresponsive property of the noncovalent charge transfer pair inside a macrocyclic host has not been investigated so far.

In this study, three cationic TPA derivatives named **G1**–**G3** (Fig. 1c) were designed and synthesized based on the mechanoresponsive behavior of TPA molecules[7]. All of **G1**–**G3** powders show a redshift of emission after grinding. The cationic **G1**–**G3** are able to bind to CB[8] to form 1:2 homoternary inclusion complexes through host-stabilized charge transfer (HSCT) interaction, and the obtained complexes exhibit a redshift of emission upon grinding (Fig. 1a) even though they are amorphous. The planarization of molecular conformation induced by grinding is responsible for the red-shifted emission of homoternary complexes. Additionally, a heteroternary complex formed between CB[8] and a D-A pair also shows MCL characteristic (Fig. 1b).

## Results

### Mechanochromic property of guests

The synthesis and characterization of new compounds are presented in the supporting information (Supplementary Figs. 1–19). Compound **G1** exhibits a higher $\Phi_F$ value (8.6%) and shorter emission wavelength (560 nm) in dichloromethane, and a significantly reduced $\Phi_F$ value (0.3%) with longer emission wavelength (611 nm) in acetonitrile (Supplementary Fig. 20). These observations suggest its twisted intramolecular charge transfer (TICT) nature[49]. The luminescent behaviors and MCL properties of **G1**–**G3** in the solid state were investigated next. As shown in Fig. 2a, **G1** is highly emissive in the solid state, exhibiting yellow fluorescence at 566 nm with an $\Phi_F$ of 32%, and the maximum emission wavelength is red-shifted to 578 nm ($\Phi_F$ = 22%) after grinding. Powder X-ray diffraction (PXRD) results disclose that the sharp and intense diffraction peaks of **G1** powder are significantly weakened by grinding (Supplementary Fig. 21b), suggesting that grinding could

partially destroy the ordered packing arrangement. Fuming with water leads to the return of its fluorescence to the initial state, accompanied by the PXRD pattern of the fumed sample being identical to the initial samples, revealing the recovery of its crystalline structure (Supplementary Fig. 21). **G2** powder shows similar spectral and PXRD pattern changes upon grinding and fuming: the emission wavelength is slightly red-shifted from 547 nm to 556 nm by grinding and restores to the initial state via fuming with water; weakened diffraction peaks after grinding restore as that of the initial sample upon fuming (Fig. 2b and Supplementary Fig. 22). **G3** emits yellow fluorescence with the maximum emission wavelength at 560 nm ($\Phi_F$ = 28%). After grinding, the emission wavelength is red-shifted to 596 (Fig. 2c and Supplementary Fig. 23a) with a decreased $\Phi_F$ value of 19%, accompanied by the emission color changing to orange. Subsequent treating the ground sample with water makes the blueshift of emission wavelength to 571 nm (Supplementary Fig. 23a). Compared to the initial powder of **G3**, the PXRD results of the fumed sample show that the diffraction peaks at 9.96°, 14.14°, and 15.61° disappear, accompanied by the observation of new intense signals at 8.93°, 11.90°, 18.43°, and 29.92° (Supplementary Fig. 23b), implying that a new crystalline state could form when the amorphous ground sample is treated with water.

As presented above, all of **G1**–**G3** exhibit MCL characteristic: the red-shifted emission is observed upon grinding, along with a decrease in fluorescence quantum yield (Table 1). These mechanoresponsive behaviors in the solid state are highly related to phase transition.

### Host–guest complexation between CB[8] and guests

Firstly, $^1$H NMR titration experiments were conducted to verify the binding mode (Fig. 3a). Upon gradually increasing the equiv. of CB[8], the signals of $H_{7-11}$ on the 4-phenylpyridine group of **G1** undergo upfield shifts ($\Delta\delta$ = −1.30 ppm, −1.16 ppm, −0.99 ppm, −1.07 and −0.30 ppm, respectively), while the signal of $H_1$ on aldehyde undergoes downfield shift ($\Delta\delta$ = 0.17 ppm), indicating that the 4-phenylpyridine group is located inside the cavity, while the aldehyde group is near the outside of the cavity. Besides, both the proton signals of bound and free **G1** are observed when the equiv. of CB[8] is 0.25, revealing that the recognition process displays slow kinetics of exchange on the $^1$H NMR time scale. Electrospray ionization mass spectroscopy (ESI-MS) test confirms the formation of the CB[8]·**G1**$_2$ complex in aqueous solution (Fig. 3b): the ion at $m/z$ = 1029.355 corresponds to the 1:2 complex ([CB[8] + 2**G1**]$^{2+}$ = 1029.362). Taking into account the strong electrostatic repulsion of pyridinium units and the lack of observation of extra split signals of CB[8] in $^1$H NMR spectra, it can be concluded that the 4-phenylpyridine groups of two **G1** should arrange in an anti-parallel manner in the cavity of CB[8][50]. As shown in Fig. 3c, the absorbance of **G1** at 251 and 353 nm decrease gradually upon adding CB[8] into the aqueous solution, and the absorbance at 415 nm exhibits an apparent bathochromic shift by 23 nm upon continuous addition of CB[8] until 0.5 equiv. This spectral change is similar to the related host–guest systems involving the formation of CT dimer in the cavity of CB[8] driven by the HSCT interaction[51,52]. Furthermore, the negligible fluorescence of **G1** in diluted aqueous solution with the maximum emission wavelength at 601 nm ($\Phi_F$ ‹ 0.1%) is greatly enhanced upon addition of CB[8] (Fig. 3d), with an $\Phi_F$ of 0.71% for CB[8]·**G1**$_2$. This enhancement can be attributed to the restricted intramolecular rotation of the chromophore (AIE effect) by the confinement effect from the nonpolar cavity of CB[8] and the weakened TICT interaction caused by complexation[49]. Also, **G1** shows an obvious increased fluorescence lifetime upon binding with CB[8]. The results of fluorescence lifetime are shown in supplementary information (Supplementary Figs. 24, 25), and the photophysical data are summarized in Table 1. Subsequently, isothermal titration calorimetry (ITC) experiment was carried out to verify the binding affinity of CB[8] towards **G1**. The data is well fitted by using a sequential binding model, showing a host–guest stoichiometry of 1:2 (Fig. 3e). According to the thermodynamic data obtained from

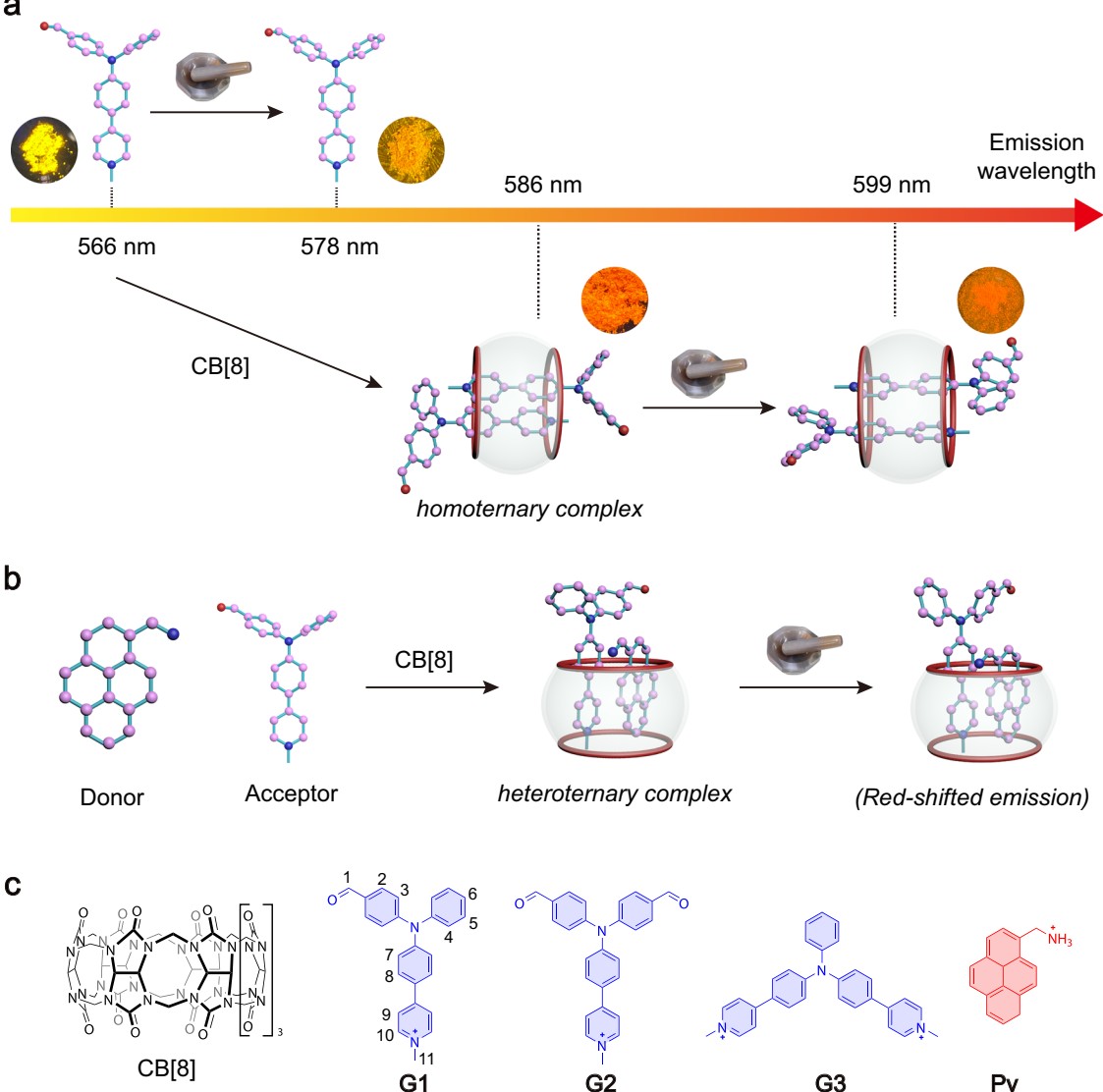

**Fig. 1 | The proposed host–guest inclusion strategy for the construction of sequentially red-shifted mechanochromic system. a** Schematic of the construction of sequentially red-shifted mechanochromic system. **b** Schematic of the formation of heteroternary inclusion complex. **c** Chemical structures of macrocyclic host CB[8] and guests (the counterions Cl⁻ are omitted for clarity).

ITC, the host–guest complexation is mainly governed by enthalpy (Supplementary Table 1). The resulted binding constant of CB[8]·**G1**₂ is as high as $7.98 \times 10^{12}\,M^{-2}$, indicating the high stability of this homoternary complex.

Afterwards, we used the same methods to investigate the binding mode between CB[8] and **G2**. NMR titration indicates that, similar to **G1**, the binding of **G2** with CB[8] exhibits slow kinetics of exchange on the ¹H NMR time scale, and two included **G2** molecules should be arranged in an anti-parallel orientation in the CB[8] cavity (Supplementary Fig. 26). The ESI-MS result (Supplementary Fig. 27) also discloses the ion at $m/z = 1057.355$ which corresponds to the 1:2 inclusion complex ([CB[8] + 2**G2**]²⁺ = 1057.361). Binding of **G2** with CB[8] leads to a redshift of its absorbance from 388 nm to 427 nm, and 65-fold/6-fold enhancement of its fluorescence lifetime/quantum yield (Supplementary Fig. 28). The two cation sites endow **G3** with better water solubility and may result in peculiar binding mode towards CB[8]. ¹H NMR titration spectra demonstrate slow-exchange kinetics for the complexation between CB[8] and **G3** (Supplementary Fig. 29). Considering the electrostatic repulsion and steric hindrance of the TPA group, we infer that only one of the 4-phenylpyridine groups for each **G3** is

located inside the cavity, and two included **G3** molecules are also arranged in a head-to-tail manner, thus resulting in the observation of two sets of proton signals of 4-phenylpyridine moiety. As shown in Supplementary Fig. 30, the absorbance of **G3** at 437 nm in aqueous solution shows an apparent redshift by 31 nm, and the emission is greatly enhanced upon binding with CB[8], along with the enhancement of fluorescence lifetime and quantum yield (Table 1). The above phenomena give a clear picture on the host–guest binding of **G1**–**G3** with CB[8]: all three guests bind with CB[8] to form stable 1:2 host–guest complexes, and the CB[8]-mediated homoternary complexes in aqueous solution exhibit longer fluorescence lifetime and higher fluorescence quantum yields than that of free guests (Table 1).

## Mechanochromic behavior of CB[8]·**G2**

Based on the above results of host–guest binding, we investigated the MCL properties of host–guest complexes. All of the host–guest complex powders were prepared by lyophilization since lyophilization is an effective way to obtain the solid sample of host–guest complex. As shown in Fig. 2a, the CB[8]·**G1**₂ powder exhibits orange fluorescence under 365 nm UV light excitation, with the maximum emission

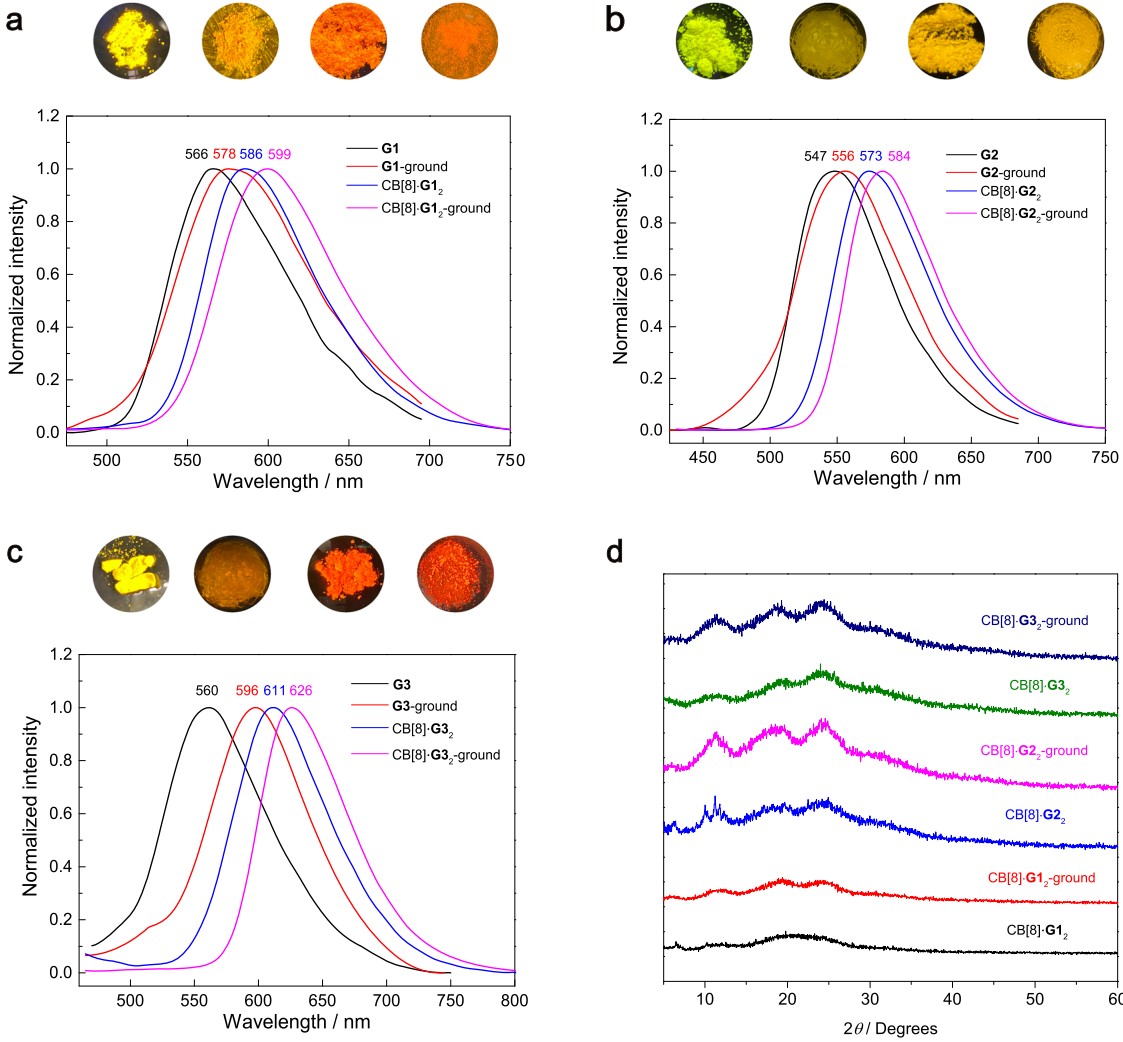

**Fig. 2 | The MCL behaviors of guests and homoternary complexes.** Photoluminescence spectra for the as-prepared and ground samples of **G1** and CB[8]·**G1**₂ (**a**), **G2** and CB[8]·**G2**₂ (**b**), **G3** and CB[8]·**G3**₂ (**c**) ($\lambda_{ex}$ = 420 nm) (insert: photographs for the as-prepared and ground samples of **G** and CB[8]·**G**₂ under 365 nm UV light excitation). **d** Powder X-ray diffraction (PXRD) patterns for the as-prepared and ground samples of CB[8]·**G**₂.

**Table 1 | Photoluminescence data of G1–G3 and their host–guest complexes in different states at room temperature**

| Compound | In solution[a] | | | In solid state | | | | | |
|---|---|---|---|---|---|---|---|---|---|
| | | | | Initial state | | | Ground state | | |
| | λ (nm)[b] | τ (ns) | Φ (%) | λ (nm)[b] | τ (ns) | Φ (%) | λ (nm)[b] | τ (ns) | Φ (%) |
| **G1**[c] | 601 | 0.08 | ˂0.1 | 566 | 4.17 | 32 | 578 | 4.11 | 22 |
| CB[8]·**G1**₂ | 587 | 1.45 | 0.71 | 586 | 3.99 | 20 | 599 | 7.58 | 10 |
| **G2** | 578 | 0.14 | 0.32 | 547 | 3.15 | 12 | 556 | 4.49 | 5.0 |
| CB[8]·**G2**₂ | 574 | 9.16 | 2.0 | 573 | 5.06 | 10 | 584 | 4.57 | 10 |
| **G3** | 608 | 0.13 | ˂0.1 | 560 | 2.42 | 28 | 596 | 2.89 | 19 |
| CB[8]·**G3**₂ | 596 | 7.39 | 2.1 | 611 | 3.25 | 12 | 626 | 3.87 | 8.3 |
| CB[8]·**Py**·**G1** | 560 | 0.42 | 4.8 | 569 | 4.43 | 18 | 583 | 4.58 | 18 |

[a]Photoluminescence data in solution are measured in water.
[b]Maximum emission wavelength.
[c]The counterions of all guests and their complexes are chloride.

wavelength at 586 nm ($\Phi_F$ = 20%). It should be noted that the CB[8]·**G1**₂ powder has a longer emission wavelength and lower $\Phi_F$ compared to free **G1**. Strikingly, the emission wavelength of CB[8]·**G1**₂ is further red-shifted to 599 nm with a reduced $\Phi_F$ of 10% after grinding. Both solid

samples of **G1** and CB[8]·**G1**₂ show a sequentially red-shifted emission upon grinding (566, 578, 586 and 599 nm represent the emission wavelength of **G1** and CB[8]·**G1**₂ powder before and after grinding, respectively), along with sequentially decreased fluorescence quantum

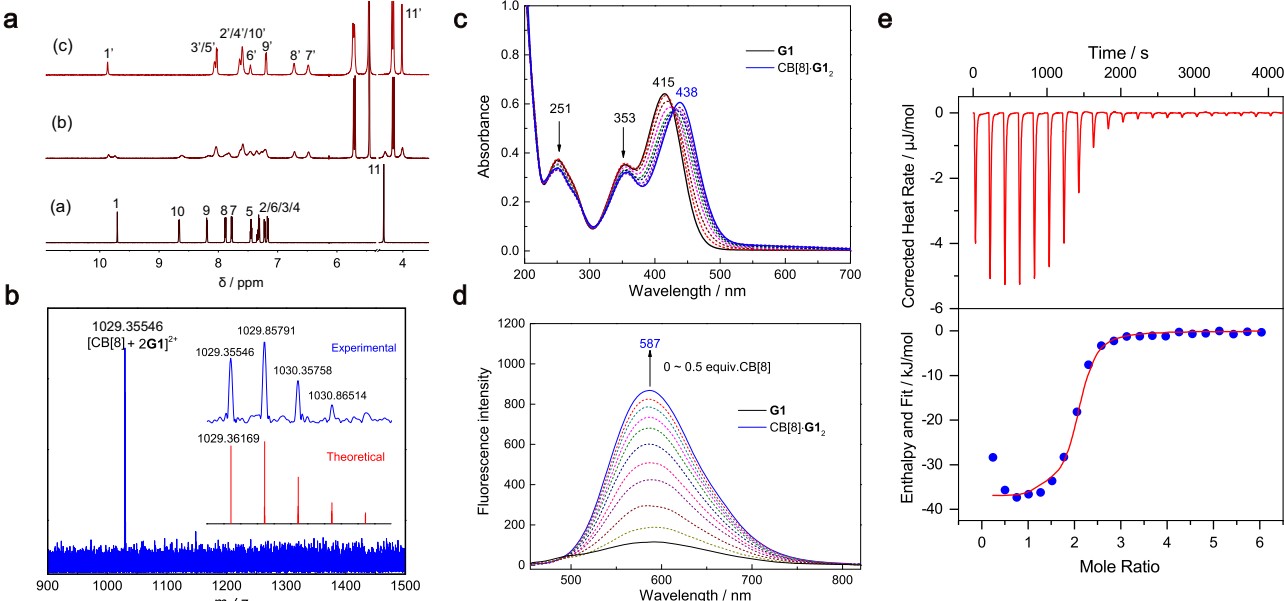

**Fig. 3 | Host–guest complexation of CB[8] with G1. a** $^1$H NMR titration spectra of **G1** ($2.0 \times 10^{-3}$ M, 600 MHz, D$_2$O, 298 K) with (a) 0 equiv., (b) 0.25 equiv., (c) 0.5 equiv. of CB[8]. **b** ESI-HRMS spectrum of CB[8]·**G1**$_2$. UV-Visible absorption (**c**) and photoluminescence spectra (**d**) of **G1** ($2.0 \times 10^{-5}$ M in H$_2$O) with different amount of CB[8] (0-0.5 equiv.) at 298 K ($\lambda_{ex} = 420$ nm). **e** ITC data for **G1** with CB[8] in H$_2$O ([**G1**] (syringe) = $2.0 \times 10^{-3}$ M, [CB[8]] (cell) = $8.0 \times 10^{-5}$ M, 298 K).

yield (32%, 22%, 20% and 10% represent the quantum yield of **G1** and CB[8]·**G1**$_2$ powder before and after grinding, respectively). Fuming with water leads to the recovery of the emission of the ground CB[8]·**G1**$_2$ powder to its initial state (Supplementary Fig. 31a). PXRD results show that all the CB[8]·**G1**$_2$ powders, with different treatments, are in an amorphous state, indicating that the MCL process of CB[8]·**G1**$_2$ is not related to the phase transition (Fig. 2d and Supplementary Fig 31b). The CB[8]·**G2**$_2$ powder also emits bright yellow fluorescence (573 nm) under UV light excitation with an $\Phi_F$ of 10%, and its emission wavelength is red-shifted to 584 nm after grinding (Fig. 2b). Both the initial and ground powder of CB[8]·**G2**$_2$ do not show observable diffraction peaks (Fig. 2d). While after the ground powder is treated with water, the emission wavelength is blue-shifted to 545 nm. The observation of many sharp signals in PXRD patterns indicates that the fumed sample is in crystalline state or partially crystalline state (Supplementary Fig. 32), hence the blue-shifted emission of the fumed sample could be ascribed to the more twisted conformation of crystalline sample than the amorphous one[53,54]. Particularly, the CB[8]·**G3**$_2$ powder shows a redshift of emission by 51 nm compared to the free **G3** powder, and its maximum emission wavelength is further red-shifted to 626 nm after grinding (Fig. 2c). Subsequent treating the ground sample with water leads to the recovery of initial fluorescence (Supplementary Fig. 33).

In view of the unique 1:2 binding mode and different photophysical properties of **G** and CB[8]·**G2**, it is reasonable to suggest that the decreased $\Phi_F$ and longer emission wavelengths of **G** upon complexation in solid state could be owing to the destruction of aggregation of free **G** and the occurrence of HSCT interaction. Moreover, due to the twisted nature of **G** and the confinement effect of CB[8], a conformational change may also occur during the host–guest complexation and further affect the luminescent behavior. Notably, all of the PXRD patterns of CB[8]·**G2** before and after grinding do not show any clear and intense diffraction peaks (Fig. 2d), indicating that grinding does not alter the amorphous state of initial sample. Based on the twisted geometry and the location of TPA moiety (outside the cavity) in the complex, we speculate that the conformation variation may also play a pivotal role in the MCL process of host–guest complex.

## Analysis of single crystals

Single crystals suitable for X-ray structural analysis of **G1** and CB[8]·**G1**$_2$ were successfully obtained, allowing us to directly explore the effect of conformation/packing on their photophysical properties and MCL mechanism. Compound **G1** crystallized in a triclinic structure with a space group of P-1. The planarity of the twisted molecule can be quantitatively characterized by the dihedral angles of adjacent (hetero)aryl groups[55]. As illustrated in Fig. 4a, there are two slightly different conformations, **G1**-a and **G1**-b, in the **G1** crystal with the sum of three dihedral angles of 187.43° (**G1**-a) and 184.13° (**G1**-b), demonstrating the highly twisted conformation of **G1**. Intermolecular interactions including C−H⋯O (2.441 Å), C−H⋯π (3.268 Å), and C−H⋯Cl (3 types, 2.801–2.948 Å) are observed (Fig. 4b). Notably, no π−π interaction is observed in the **G1** crystal due to the twisted geometry of **G1**, which blocks the overlap of the π-plane. Since the loose packing of the crystal induced by the steric hindrance and twisted conformation can be readily amorphized under mechanical stimulation[56], the observation of MCL behavior of **G1** is reasonable. Upon amorphization by grinding, the high twisting stress and weak interactions are destroyed, generating a relatively planarized conformation, which is accompanied by a decreased quantum yield and a redder emission (Table 1).

In agreement with the conclusions based on the $^1$H NMR titration experiments, the crystal structure of CB[8]·**G1**$_2$ directly shows that the 4-phenylpyridine units of two **G1** are located inside the cavity of CB[8] in a head-to-tail manner (Fig. 4d, e). There are four different conformations of **G1** (named **G1**-c, d, e, f, respectively) in the CB[8]·**G1**$_2$ crystal, and two **G1** with similar conformations are paired to form two types of homoternary host–guest complexes (Fig. 4d, f). Intriguingly, the sum of the three dihedral angles for **G1**-c, d, e, f are smaller than those of free **G1**, indicating that **G1** in the inclusion complex is less twisted (Supplementary Table 2). Especially for **G1**-c and **G1**-d, the 4-phenylpyridine groups are almost coplanar, with dihedral angles of 5.09° and 7.77°, respectively (Fig. 4g). The data of the dihedral angles in **G1**-involved crystals are collected in Supplementary Table 2 for clear comparison. Crucially, since two 4-phenylpyridine groups of two **G1** molecules are located inside the confined cavity, π−π interactions are observed in the CB[8]·**G1**$_2$ crystal. The centroid-centroid distances

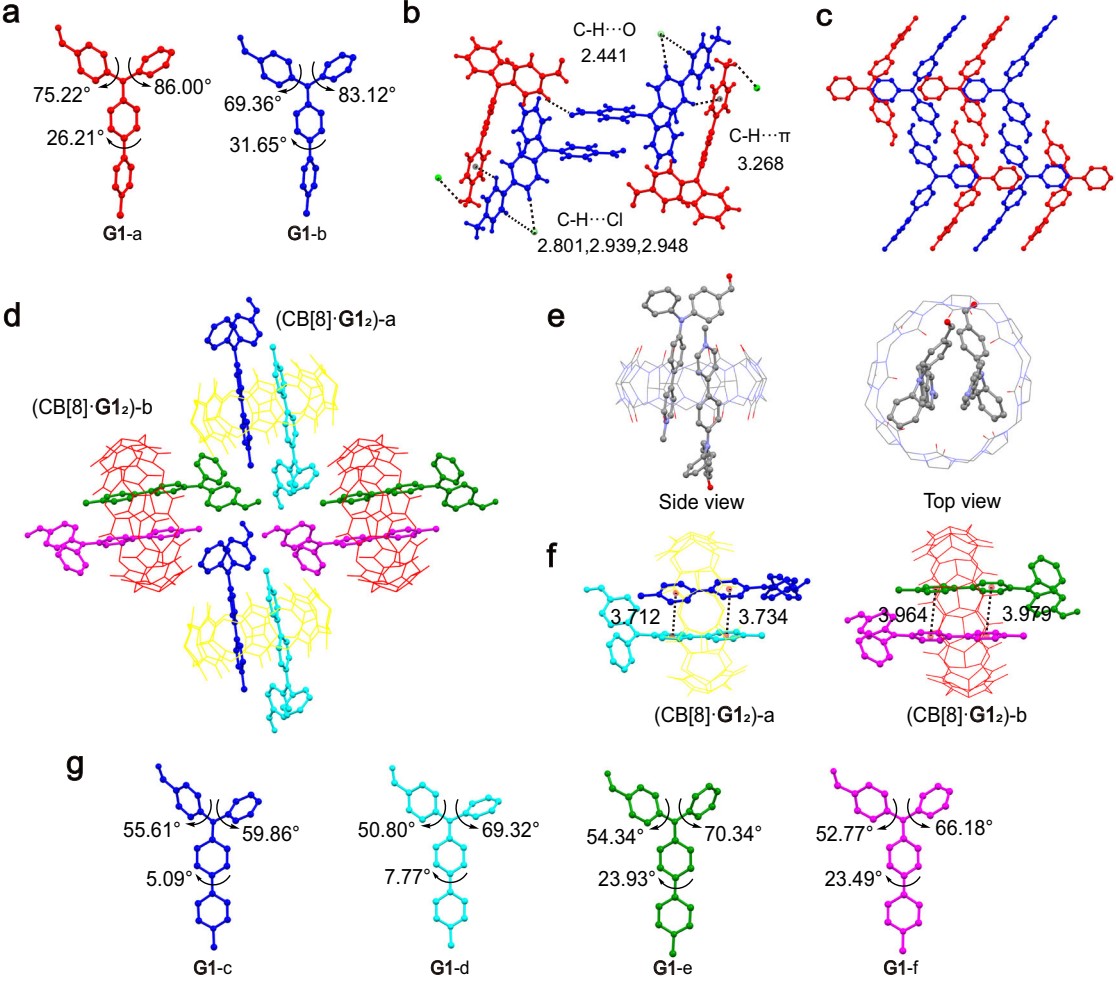

**Fig. 4 | Crystal structure of G1 and CB[8]·G1₂.** **a** Two conformations of **G1** (**G1**-a and **G1**-b). **b** Intermolecular C−H···O, C−H···Cl and C−H···π interactions in **G1** crystal. **c** The packing mode of **G1** (view along a-axis). **d** Molecular arrangement of CB[8]·**G1₂** in crystal (view along b-axis). **e** Side view and top view of CB[8]·**G1₂**-a. **f** The π–π interactions of **G1** in the cavity of CB[8] (the centroid-centroid distances between the neighboring benzene and pyridine rings are presented). **g** Conformations of **G1** in the CB[8]·**G1₂** complex (**G1**-c, **G1**-d, **G1**-e, and **G1**-f). Hydrogen atoms and counterions are omitted in a and c–g for clarity.

between the neighboring benzene and pyridine groups range from 3.712 Å to 3.979 Å (Fig. 4f). According to the results presented above, the host–guest complexation not only arranges the conformation of **G1** in a more planar state but also enhances the π–π interaction due to the unique binding mode, which is responsible for the red-shifted emission of CB[8]·**G1₂** compared to free **G1** in solid state. In addition, the strong ion–dipole interactions (Supplementary Fig. 34) and hydrophobic effect are synergistically to stabilize the ternary host–guest complex. And each CB[8]·**G1₂** interacts with adjacent CB[8] through multiple C−H···O interactions (Supplementary Fig. 35).

Single crystals of CB[8]·**G2₂** and CB[8]·**G3₂** were also obtained by slowly evaporating their aqueous solution at room temperature. Similar to **G1**, the 4-phenylpyridine groups of **G2** adopt an anti-parallel orientation inside the cavity of CB[8], and two types of inclusion complexes are formed by pairs of two different conformations of **G2** (Fig. 5a, c). The benzene rings of two **G2** inside the cavity are perfectly parallel with interplanar distances of 3.419 (**G2**-a) and 3.245 Å (**G2**-b), respectively

(Fig. 5d, e). The distances between the centroid of pyridine and the plane of benzene on adjacent **G2** molecules in two inclusion complexes are 3.352 (**G2**-a) and 3.530 Å (**G2**-b), respectively (Fig. 5d, e). These close distances reveal strong π–π interactions of the **G2** dimer inside the cavity. The crystal structure of CB[8]·**G3₂** shows that one of the 4-phenylpyridine groups on each **G3** is encapsulated by CB[8], and the two different conformations are paired to form two types of ternary complex, respectively (Fig. 5b, f). Also, the close stacking of **G3** molecules inside the cavity implies strong π–π interactions (Fig. 5g, h). The main crystallographic parameters are summarized in Supplementary Tables 3–5. Although, besides the CB[8]·**G3₂** complex, there are two cavity-free CB[8] molecules that interact with neighboring **G3** through ion–dipole and C−H···O interactions in each unit cell, this crystal structure still sheds light on the analysis of the host–guest binding mode and the conformation of **G3** in the inclusion complex.

The X-ray crystallographic analysis mentioned above indicates that although the 1:2 anti-parallel binding makes the guests adopt a

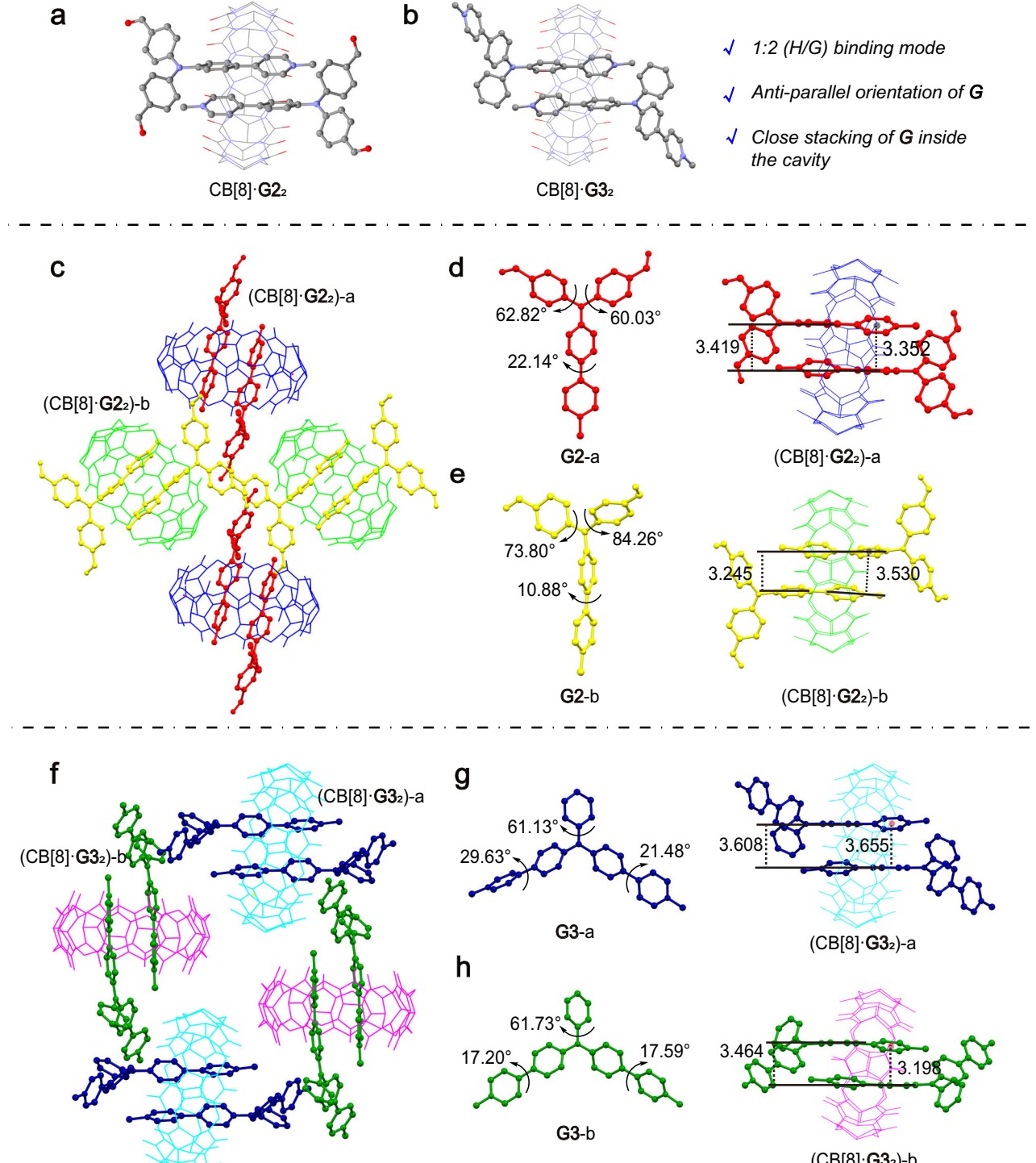

**Fig. 5 | Crystal structure of CB[8]·G2₂ and CB[8]·G3₂. a** Overview of the binding mode of CB[8]·G2₂. **b** Overview of the binding mode of CB[8]·G3₂. **c** Unit cell of CB[8]·G2₂ (view along a-axis). **d** Conformation of G2-a and π–π interactions of G2-a inside the cavity. **e** Conformation of G2-b and π–π interactions of G2-b inside the cavity (the distances between benzene ring of two G2 in the cavity, between the centroid of pyridine and the plane of benzene on adjacent G2 in the cavity are presented). **f** Partial molecular packing of CB[8]·G3₂. **g** Conformation of G3-a and π–π interactions of G3-a inside the cavity. **h** Conformation of G3-b and π–π interactions of G3-b inside the cavity (the distances between benzene ring of two G3 in the cavity, between the centroid of pyridine and the plane of benzene on adjacent G3 in the cavity are presented). Hydrogen atoms and counterions are omitted for clarity.

more planar state, the TPA moieties still exhibit twisted conformation in the inclusion complex. Although the powders of host–guest complexes in their amorphous states do not show regular packing as crystals, each inclusion complex maintains its specific binding mode at molecule level to form the macroscopic amorphous powder. The introduction of the host–guest inclusion strategy in producing this type of amorphous organic MCL system may be an enlightening principle in the design of future MCL materials, as it could bypass the complicated and demanding conditions for the preparation of crystalline MCL samples.

Considering that the mechanochromic process of the ternary complex does not involve phase transition, and part of the twisted TPA

moiety located outside the cavity is still sensitive to the mechanical stimulation, we believe the planar conformation induced by grinding is the most reasonable mechanism. As changes in molecular conformation often lead to changes in absorption peaks[57], solid state absorption spectra of **G1** and CB[8]·**G1**$_2$ in different states were performed to validate our speculation. As shown in Supplementary Fig. 36, both **G1** and CB[8]·**G1**$_2$ exhibit obvious red-shifted absorbances after grinding, confirming that the alternated molecular conformation is the key origin of the MCL behavior. The host–guest complexation enables the encapsulated guests to achieve extra red-shifted luminescence upon grinding by introducing HSCT interaction.

### Formation of heteroternary complex CB[8]·Py·G1 and its mechanochromic behavior

Since the CT interaction between donor and acceptor molecules can be enhanced by forming hetero-guest pairs in the cavity of CB[8][48], we continued to explore whether the host–guest inclusion strategy is applicable to hetero-guest pairs for the construction of MCL system. 1-Pyrenemethylamine hydrochloride (**Py**-Cl, chlorine is omitted for clarity), retaining electron-rich nature of the pyrene unit and showing adequate solubility in water, was chosen as an ideal electron-donor molecule to construct a heteroternary complex. The binding of **Py** with CB[8] was investigated using $^1$H NMR titration spectroscopy (Supplementary Fig. 37). Upon addition of CB[8] to the solution of **Py**, all proton signals of the pyrene moiety show upfield shifts, demonstrating the encapsulation of the pyrene moiety inside the cavity. While the signal of methylene protons exhibits a slight downfield shift, from 4.88 to 5.08 ppm, suggesting that the NH$_3^+$ group is outside but near the carbonyl portal of CB[8]. Additionally, the absorbance and emission in diluted aqueous solution decrease upon addition of CB[8] (Supplementary Fig. 38). These results are consistent with a previous report that the cavity of CB[8] can accommodate one pyrene group[58].

The formation of heteroternary complex was then checked by $^1$H NMR titration, UV/vis, and fluorescence spectroscopies in aqueous solution. As shown in Supplementary Fig. 39a–c, the proton signals of **G1** slightly shift when **G1** is mixed with **Py** in equal molar ratio. Intriguingly, new proton signals that are different from CB[8]·**Py** and CB[8]·**G1**$_2$ are observed in the $^1$H NMR spectrum of the 1:1:1 mixture of CB[8], **Py**, and **G1**, confirming the formation of the CB[8]·**Py**·**G1** complex. Although the recognition process of CB[8] towards the hetero-guest pair displays fast exchange kinetics on the $^1$H NMR time scale (Supplementary Fig. 40), based on the fact that the binding between CB[8] and **G1** displays slow-exchange kinetics and no proton signals of CB[8]·**G1**$_2$ are found in the titration NMR spectra, we highly believe that the heteroternary inclusion complex is exclusively formed in aqueous solution. ESI-MS also provides strong evidence for the formation of the heteroternary complex. In the ESI-MS of an equimolar mixture of **G1**, **Py**, and CB[8], the ion at $m/z = 962.835$ corresponding to the 1:1:1 complex ([CB[8] + **Py** + **G1**]$^{2+}$ = 962.835) is observed (Supplementary Fig. 41). Upon gradually adding CB[8] to the aqueous solution of 1:1 mixture of **Py** and **G1**, the absorbances at 241, 275 and 340 nm decrease, along with a redshift of absorbance from 414 nm to 437 nm (Supplementary Fig. 42). Due to the strong blue emission of **Py** in diluted solution ($2.0 \times 10^{-5}$ M), the weak emission signal of **G1** is hardly visible in the 1:1 mixture of **Py** and **G1** under the same experimental condition (Fig. 6d). At high concentration ($1.0 \times 10^{-3}$ M), the mixture of **Py** and **G1** shows a weak emission band at 514 nm with a broad full width at half maxima and a long lifetime ($\tau = 37.97$ ns) (Supplementary Figs. 43, 44), which could be ascribed to the formation of the exciplex **Py·G1**[59]. A broad emission band at 560 nm emerges and is enhanced by gradually adding CB[8] (from 0 to 1.0 equiv.) to the 1:1 mixture of **Py** and **G1** even in diluted aqueous solution (Fig. 6d). The excitation spectrum of CB[8]·**Py**·**G1**, collected by monitoring the emission at 560 nm, shows a maximum at about 490 nm, which closely matches with the corresponding charge transfer absorption band of

CB[8]·**Py**·**G1** (Supplementary Fig. 44). Therefore, the newly generated yellow emission at 560 nm could originate from the HSCT interaction. We believe that the heteroternary complexation narrows the distance and corrects the orientation between **Py** and **G1**, resulting in the stronger charge transfer interaction between **Py** and **G1**[44].

The cube-shaped yellow crystals of CB[8]·**Py**·**G1** were successfully obtained by slow evaporation of the aqueous solution (1:1:1 mixture of CB[8], **Py**, and **G1**) at room temperature over a period of one month. It is worth mentioning that the CB[8]·**Py**·**G1** crystal provides another X-ray crystallography evidence for heteroternary complex, following the report by Kim's group on the first CB-based hetero-guest pair crystal in 2001[48]. As shown in Fig. 6a, two types of CB[8]·**Py**·**G1** (designated as a and b, respectively) with different binding sites are arranged alternately in each unit cell. In (CB[8]·**Py**·**G1**)-a, the pyrene group and methylated pyridinium ring of **G1** are located inside the cavity, and two guests are arranged in a head-to-tail manner (Fig. 6b). The pyrene plane is almost perpendicular to the equatorial plane of CB[8] with a dihedral angle of 88.93°. **G1**-g shows smaller dihedral angles of its adjacent (hetero)aryl groups than **G1**-a/b, suggesting that the heteroternary binding style also results in a less twisted geometry of **G1** (Supplementary Table 2). The methylated pyridinium ring of **G1** and pyrene moiety of **Py** are almost parallel, showing a dihedral angle of 1.14° and a mean separation of 3.287 Å (Fig. 6b). These values are typical for the charge transfer pair inside CB[8] in Kim's work (additionally, the arrangement of pyridinium and π-moiety of pyrene in the cavity also fits the category of cation–π interaction[60]). In (CB[8]·**Py**·**G1**)-b, the dihedral angle of pyrene plane and the equatorial plane of CB[8] is 51.73°, and two guests are arranged in a head-to-head manner (Fig. 6c). Similarly, **G1**-h also shows a less twisted conformation than free **G1**, and the dihedral angle between methylated pyridinium ring of **G1** and pyrene moiety is 4.71°. Besides the HSCT interaction, the stronger ion–dipole interactions between guests and C=O of CB[8] are also responsible for the stability of CB[8]·**Py**·**G1** complex (Supplementary Fig. 45). Additionally, a large amount of intermolecular C–H···O interactions between adjacent inclusion complexes are observed (Supplementary Fig. 46).

With the observation of a bright yellow emission induced by HSCT interaction in aqueous solution, we decided to explore whether the HSCT interaction could affect the MCL behavior of the D-A system. Before beginning our investigation, the luminescent property of a 1:1 mixture of **Py** and **G1** (the solid sample was prepared by lyophilization of a homogeneous solution of 1:1 mixture of **Py** and **G1**) was examined. The powder shows slightly red-shifted absorption and emission compared to the individual **Py** and **G1** (Supplementary Fig. 47), suggesting the existence of a D-A structure[61]. However, grinding the sample leads to a negligible redshift of emission by 4 nm (Supplementary Fig. 48), suggesting that the mechanical stimulation could not be able to alter the arrangement of this D-A structure. In contrast, the CB[8]·**Py**·**G1** powder emits yellow fluorescence at 569 nm with an $\Phi_F$ of 18% (Fig. 6e). After grinding, the emission is red-shifted to 583 nm, and treating the ground powder with water restores the emission wavelength to its initial state. Only few weak diffraction signals are observed in the PXRD pattern of CB[8]·**Py**·**G1** powder, and grinding transforms the morphology of the sample into an amorphous state (Supplementary Fig. 49). The weak diffraction signals reappear after the ground powder is treated with water. Grinding also leads to a red-shifted absorption of the CB[8]·**Py**·**G1** powder (Supplementary Fig. 50). As **Py** and 4-phenylpyridine of **G1** are located inside the confined cavity, the mechanical stimulation is more likely to cause a conformation planarization of the unencapsulated TPA groups and subsequently induce stronger CT interaction, resulting in the red-shifted emission of the heteroternary complex. Theoretical calculations were then conducted to gain a deeper insight into the MCL behavior of the heteroternary complex. As shown in Supplementary Fig. 51, two optimized structures

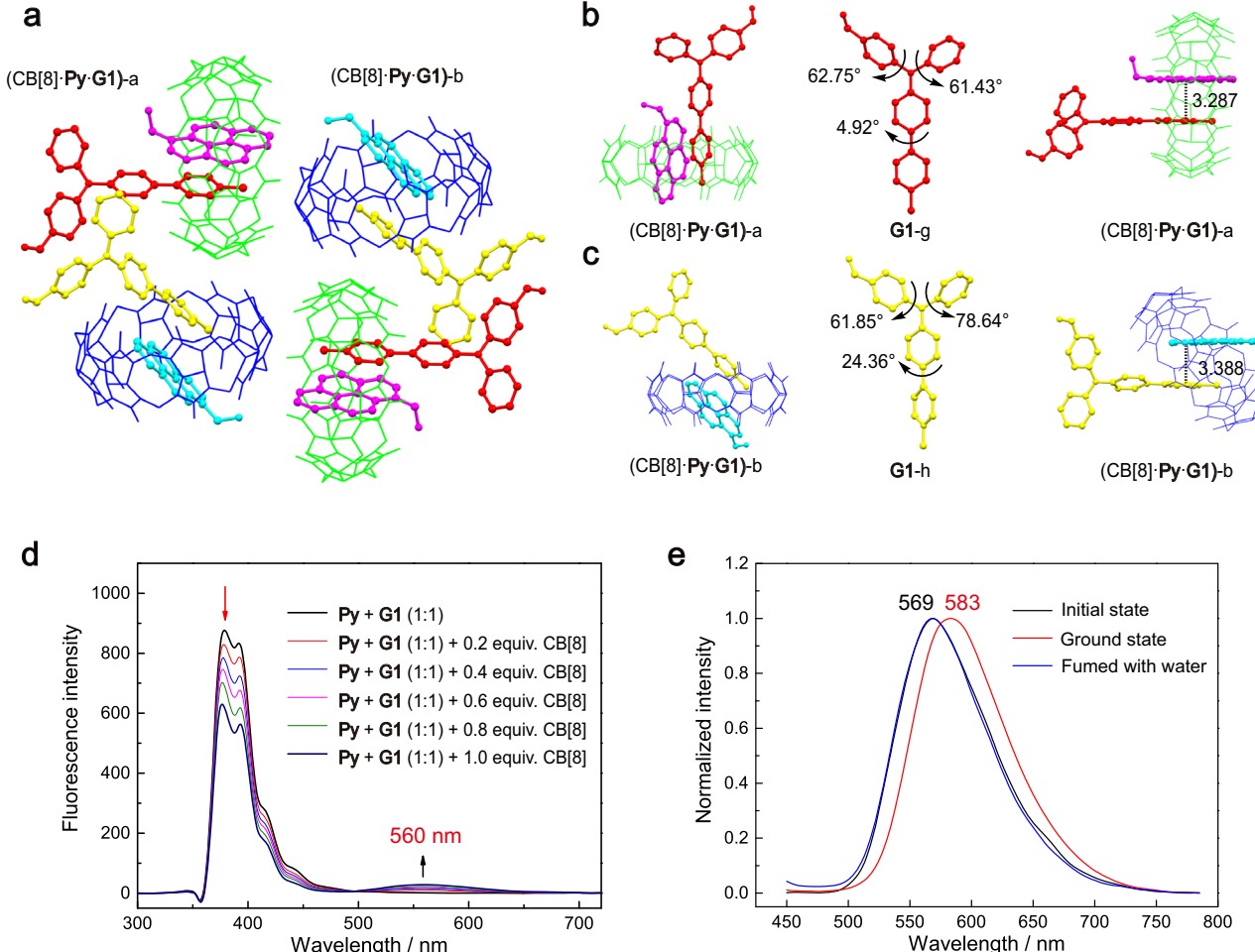

**Fig. 6 | Crystal structure and MCL behavior of the heteroternary complex CB[8]·Py·G1. a** Unit cell of CB[8]·Py·G1 crystal (two types of heteroternary complex are named (CB[8]·Py·G1)-a and (CB[8]·Py·G1)-b, respectively). **b** Binding mode of (CB[8]·Py·G1)-a, conformation of **G1**-g and the distance between the pyrene plane of **Py** and centroid of pyridinium on **G1** in (CB[8]·Py·G1)-a. **c** Binding mode of (CB[8]·Py·G1)-b, conformation of **G1**-h and the distance between the pyrene plane

of **Py** and centroid of pyridinium on **G1** in (CB[8]·Py·G1)-b (hydrogen atoms and counterions are omitted for clarity). **d** Fluorescence emission spectra of **Py** + **G1** (1:1, $2.0 \times 10^{-5}$ M in H$_2$O, 298 K) with different equiv. of CB[8] ($\lambda_{ex}$ = 340 nm). **e** Photoluminescence spectra of CB[8]·Py·G1 powder under different treatments ($\lambda_{ex}$ = 420 nm).

of heteroternary complex, (CB[8]·Py·G1)-α and (CB[8]·Py·G1)-β, were built based on the single crystal of CB[8]·Py·G1. Among them, **G1** in (CB[8]·Py·G1)-β is more planar than that in (CB[8]·Py·G1)-α. Importantly, (CB[8]·Py·G1)-β with a planar **G1** exhibits a red-shifted emission compared to (CB[8]·Py·G1)-α with a more twisted **G1**, suggesting that the planar conformation of **G1** may be responsible for the red-shifted emission under mechanical stimuli. This calculation result is in line with the experimental observation of the red-shifted emission of the heteroternary complex CB[8]·Py·G1 upon grinding[62]. The independent gradient model (IGM) analysis[63] for (CB[8]·Py·G1)-α/β clearly reveals the existence of multiple noncovalent interactions (the green region) between the host and guests (Supplementary Fig. 52), which is in agreement with the X-ray crystallographic analysis. This calculation result could also explain the red-shifted emission of the homoternary complexes upon grinding, since the TPA groups of **G1**–**G3** in the homoternary complexes are also located outside the cavity, as shown in the crystal structures. Besides, the steric effect of CB[8] may lead to a loose packing, increasing the likelihood of mechanical stimulation inducing a change in molecular conformation. Moreover, the destruction of a significant amount of intermolecular interactions upon grinding, particularly the intermolecular hydrogen bonds between adjacent inclusion complexes, could synergistically result in a larger spectral shift compared to the D-A structure without the host[64].

## Discussion

We have proposed a host–guest inclusion-directed strategy for the rational design of sequentially red-shifted mechanochromic materials. A series of TPA derivatives (**G1**–**G3**) show redshifts of emission upon grinding owing to planarization of TPA unit under mechanical stimulation. These three guests bind with CB[8] host in a 1:2 stoichiometry to form stable homoternary complex CB[8]·**G2**, which is evidenced by ¹H NMR, ESI-MS, ITC, UV/vis, fluorescence spectroscopies, and X-ray crystallography. The host–guest complexes display bathochromic-shifted emission compared to free guests in solid state, due to the host–guest inclusion inducing enhanced intermolecular charge transfer interaction and less twisted conformation of guests. Furthermore, grinding leads to a further redshift of emission of homoternary complexes, resulting from the planarization of unencapsulated moieties by mechanical stimulation. The sequentially red-shifted mechanochromic system has been successfully achieved through the host–guest inclusion strategy without complicated molecular design and synthesis, and the construction of an amorphous mechanoresponsive complex (CB[8]·**G2**) provides a more convenient procedure for the preparation of MCL systems. Additionally, the host–guest encapsulation of a donor–acceptor pair in CB[8] endows the heteroternary complex with better responsiveness toward mechanical stimulation, with a large redshift of emission. The host–guest inclusion

strategy in this work expands the spectral shift compared to free mechanoresponsive guest upon mechanical stimulation by introducing the HSCT interaction, which we believe could be an inspirational and relatively universal method for the construction of novel MCL materials.

## Methods

### Materials and characterization methods

ESI-MS spectra were recorded on a 9.4 T high-resolution FT-MS mass spectrometer (Solarix 9.4T) from Bruker in the positive polarity. Fluorescence spectra were measured on a PerkinElmer LS-55 machine at 298 K, and a Suprasil Quartz (QS) cuvette with 1 cm pathlength was used for the fluorescence spectra measurements in aqueous solution. Milli-Q water (18.2 M$\Omega$•cm) was used for preparation of all nondeuterated aqueous solutions. The UV/Vis experiments were performed on a SHIMADZU UV-36002 instrument with 1 cm pathlength cells at 298 K. All $^1$H NMR and $^{13}$C NMR spectra were collected on Agilent 600 MHz DD2 at 298 K. The absolute fluorescence quantum yields were determined by using a Hamamatsu C9920-02G Instruments Integrating Sphere Module (SC-30) on the FS5 spectrofluorometer. Fluorescence spectra analysis software Fluoracle and Quantum Yield Wizard were used to calculate quantum yields. Fluorescence quantum yield measurements were done for 1 mL aqueous solutions with an optical density in the range of 0.06–0.1. ITC experiments were carried out on a Nano ITC (TA Instruments) at 298 K (the first data point was removed from the data set prior to curve fitting; the intervals between injections were set as 200 s). The results of PXRD were collected on the Bruker D8 Advance powder-X diffractometer. The copper target provided K$\alpha$ rays ($\lambda = 1.5418$ Å). All chemical reagents were purchased commercially and used directly without further purification. The synthesis and characterization are summarized in the supplementary information. The time-resolved fluorescence decay curves were obtained on a FS5 time-correlated single-photon-counting (TCSPC) instrument.

### Crystal growth

15.0 mg **G1** was dissolved in 5.0 mL H$_2$O. After two weeks at room temperature, yellow flake crystals were precipitated out from the concentrated solution. The single crystals of CB[8]·**G$_2$** and CB[8]·**Py·G1** were provided by standing the corresponding aqueous solution at room temperature, and the concentration of guests in these samples is $2.5 \times 10^{-3}$ M.

### Solid samples preparation

(1) CB[8]·**G$_2$**: **G1** (20 mg, 0.050 mmol) was dissolved in 15 mL deionized water. After added CB[8] (33.22 mg, 0.025 mmol) to the solution, the mixture was sonicated for about 5 min until CB[8] was dissolved, and then the solid sample of CB[8]·**G1$_2$** was obtained by lyophilization. Solid samples of CB[8]·**G2$_2$** and CB[8]·**G3$_2$** were obtained by the same method. (2) CB[8]·**Py·G1**: **Py** (13.39 mg, 0.050 mmol), **G1** (20 mg, 0.050 mmol) was dissolved in 25 mL deionized water. After added CB[8] (66.44 mg, 0.050 mmol) to the solution, the mixture was sonicated for about 8 min until CB[8] was dissolved, and then the solid sample of CB[8]·**Py·G1** was obtained by lyophilization. (3) 1:1 mixture of **Py** and **G1**: **Py** (13.39 mg, 0.050 mmol) and **G1** (20 mg, 0.050 mmol) were added in 25 mL deionized water, and the mixture was sonicated for about 5 min until **Py** and **G1** was dissolved. The solid sample was obtained by lyophilization.

### Theoretical calculations

The specific process for conformational search was performed by using ORCA5.0.3 package[65]. The ground state geometry optimization of (CB[8]·**Py·G1**)-$\alpha$/$\beta$ was conducted using the B3LYP-D3/def2SVP level[66]. The first excited state geometry optimization of (CB[8]·**Py·G1**)-$\alpha$/$\beta$ was conducted by the wB97X-D3/def2SVP level[62,67]. The Visual Molecular Dynamics (VMD) software was employed for visualization[68].

## Data availability

All data needed to support the conclusions of this manuscript are provided in the main text or Supplementary Information file. CCDC 2159417 (**G1**), 2201842 (CB[8]·**G1$_2$**), 2201844 (CB[8]·**G2$_2$**), 2201854 (CB[8]·**G3$_2$**) and 2201856 (CB[8]·**Py·G1**) contain the supplementary crystallographic data for this paper. These data can be obtained free of charge via www.ccdc.cam.ac.uk/data_request/cif, or by emailing data_request@ccdc.cam.ac.uk, or by contacting The Cambridge Crystallographic Data Centre, 12 Union Road, Cambridge CB2 1EZ, UK; fax: +44 1223 336033. All data is available from the corresponding author upon request.

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

## Acknowledgements

This work was financially supported by the National Natural Science Foundation of China (21871216: S.L., 21901194: X.H.).

## Author contributions

S.L. and X.H. conceived and designed the project. D.S. conducted the synthesis and measurements, and wrote the manuscript. D.S., Y.W., X.H. and S.L. reviewed and edited the manuscript. All authors discussed the results and commented on the manuscript.

## Competing interests

The authors declare no competing interests.
