## [Peer Review File · Nature Communications]

REVIEWER COMMENTS

Reviewer #1 (Remarks to the Author):

The authors developed a series of charge-transfer (homo- and hetero-) dimers stabilized by host-guest interactions, presenting a distinct mechanochromic system. The host-guest dimers with detailed single-crystal information offer a clarified platform for the photochemistry study of dye aggregates. Significantly, this host-guest system can support distinct information of chromophore dimers comparing with the host-free and covalently modified counterparts due to the confined cavity of CB8 host. In general, I highly recommend this manuscript to be accepted by Nature Communications after revision:

1) The attribution of the quenched fluorescence of G1/Py mixture to “weak charge-transfer” emission (see the legend of Fig S31) is questionable. This phenomenon should be demonstrated in main text rather than simply mentioned in Supporting Information. My deduction of this problem is the formation of exciplex between G1 and Py after being excited. According to the Fig S30, the absorption spectrum of G1/Py lacks CT band and the maximum peaks of G1 (see Fig 2C) and G1/Py are nearly identical, indicating an absence of CT in ground state. Since the weak association of excited G1/Py, the emission of exciplex tends to be quenched. For host-guest system, CB[8] can act as an ideal confined container to stabilize G1/Py in both ground and excited states, thus showing a considerable variation in absorption and emission spectra. Relevant references such as ACS Appl. Mater. Interfaces 2019, 11, 14399–14407 may help to understand this issue further.

2) The authors should examine the emission curve of G1+Py again because neither weak charge-transfer nor exciplex could reasonably explain this spectrum. Since the emission of G1+Py is rather weak, the trace fluorescent impurities may interfere test result. Authors can compare the excited spectrum with the absorption one, if the two spectra don't fit well, the possibility of impurity cannot be ignored.

3). Authors explained the emission enhancement of G1 upon addition CB[8] that “is mainly due to the restricted intramolecular rotation of the chromophore (AIE effect) by the confinement effect from the cavity of CB[8]”. This explanation is reasonable but probably not sufficient. An additional possibility is that the low polarity of CB[8] cavity could also contribute to the planarization of G1 and weaken the intramolecular charge-transfer (such as TICT, which can extremely suppress emission) in aqueous condition. Authors can record the QY of G1 in low polar solvents such as dichloromethane to examine this issue.

4) Support theoretical calculation (such as DFT) and improve the single-crystal drawing can help this manuscript to step up a level.

5) Compared with G1 and G3, only CB[8]·G22 could not return to its initial state after fuming with water. Please explain the reason.

Reviewer #2 (Remarks to the Author):

In the manuscript, the authors described a host-guest inclusion-directed strategy to construct sequentially red-shifted mechanochromic materials. A series of TPA derivatives were generated, and then used for the MCL property comparison with their complexes stabilized with CB[8]. The report is impressive, but still need some improvement before the acceptance:

1. " G1 is highly emissive in solid state, exhibiting yellow fluorescence at 566 nm with an ΦF of 32.47%, and the maximum emission wavelength is red-shifted to 578 nm ($\Phi F = 21.64\%$) after grinding. Powder X-ray diffraction (PXRD) results disclose that the sharp and intense diffraction peaks of G1 powder is weakened significantly by grinding (Supplementary Fig. 11b), suggesting that grinding could partially destroy the ordered packing arrangement.." Actually, red-shifted emission is corresponding to the stronger packing or aggregation. What is the real relationship between the red shifted emission and the amorphous state?
2. The excitation wavelength values should be listed with the emission data.
3. Since ΦF value detection always adopts error about 10 %, too many significant digits shown in the value are meaningless. 2 significant digits are enough in most cases.
4. About the binding constant calculation shown in Fig.2, what is the expansion of the important signals shown in high resolution mass spectrum? Does it fit well with the simulated data? What are observed in fluorescence and Uv-vis titration after the molar ratio of CB[8] larger than 0.5 molar equiv.? How about the binding constants from fluorescence and Uv-vis titration? And how about the other thermodynamic parameters from ITC? The comparison of the binding constants from different methods related to same reasonable equilibriums should be listed.
5. The text should be improved. For example, line 18, "intermolecular change transfer" should be "intermolecular charge transfer". Line 19, "longer emission than that of free guests (even longer than ground guests)," should be "longer emission wavelength compared with free guests in solution, even corresponding pure guests in solid state." Line 31, "triphenylamine (TPA) and tetraphenylethylene derivatives and others" should be "triphenylamine (TPA), tetraphenylethylene derivatives, and others". And more other writing parts should be careful checked and improved.

Reviewer #3 (Remarks to the Author):

In the manuscript entitled "Title: The Host-guest Inclusion Driven by Host-stabilized Charge Transfer for Construction of Sequentially Red-shifted Mechanochromic System" the effect of host-guest interaction in influencing the mechanochromic properties of organic molecules has been explained by the authors. Authors have used a pyridine functionalized with triphenyl amine and have formed homoternary inclusion complexes with cucurbit[8]uril (CB[8]) by host stabilized intermolecular charge-transfer interaction. The molecules G1-G3, exhibited red-shifted emission upon grinding in the solid-state which is ascribed to increase in the planarity of conformation induced by grinding. The complexation with CB[8] has further red-shifted emission maxima pertaining to the host stabilized intermolecular charge-transfer interaction. Grinding of this complex has further shifted the emission towards higher wavelength. Authors have done extensive studies to characterize the complexation and further spectroscopic properties to justify the mechanochromic behavior. The manuscript has enough novelty and suitable to publish in this journal. However, the author should address the following points.

1. Authors have observed a difference in the mechanochromic properties of G3 both in solid state and in complexed state with CB[8]. For example, the PXRD pattern of solid ground G3 obtained after fuming with water is different from the PXRD pattern of G3 before grinding. Similar observations were shown by G3 complexed with CB[8] as well. How does the authors account for this difference in behaviour of G3 compared to G1 and G2.
2. "Also, G1 shows an obvious increased fluorescence lifetime upon binding with CB[8]". The lifetime decay plot is missing in both main text and SI. Authors should show the decay profile.
3. The authors should account the reason for decrement in the absorbance for π - π^* band on forming the ternary complex in all the cases.
4. Authors have assumed that since the triphenyl amine is located outside the cavity, the complex is sensitive to mechanical stimuli. Could this statement be made more convincing by elucidating in a quantitative manner or by computational studies.
5. Since there is a charge-transfer interaction between Py and G1, authors should also confirm the same in solid state by comparing the individual absorption and emission spectra of Py and G1 with Py:G1.
6. It is intriguing to see that the mixture Py:G1 is insensitive to mechanical stimuli, while the same complex inside CB[8] is mechanochromic. How does the authors justify this observation. Also, how is the lifetime of respective emission band changing from 1:1 mixture of Py:G1 to the 1:1:1 mixture of Py:G1:CB[8].
7. Authors may refer to other kinds of host stabilized charge-transfer interaction and refer them in the manuscript. J. Am. Chem. Soc. 2022, 144, 10854.
8. There are certain typos in the main text and supporting information (SI); for instance, in the main text page 10, line 182, 'eimssion' has to be replaced by 'emission', page 11, line 211, 'amphorization' has to

be replaced by 'amorphization', page 14, line 276, 'encapsated' to be replaced by 'encapsulated', and, page 17, line 277, 'ledas' to 'leads'. Authors should correct these typos.

Response to reviewers

Reviewer #1 (Remarks to the Author):

The authors developed a series of charge-transfer (homo- and hetero-) dimers stabilized by host-guest interactions, presenting a distinct mechanochromic system. The host-guest dimers with detailed single-crystal information offer a clarified platform for the photochemistry study of dye aggregates. Significantly, this host-guest system can support distinct information of chromophore dimers comparing with the host-free and covalently modified counterparts due to the confined cavity of CB8 host. In general, I highly recommend this manuscript to be accepted by Nature Communications after revision:

1. The attribution of the quenched fluorescence of **G1/Py** mixture to “weak charge-transfer” emission (see the legend of Fig S31) is questionable. This phenomenon should be demonstrated in main text rather than simply mentioned in Supporting Information. My deduction of this problem is the formation of exciplex between **G1** and **Py** after being excited. According to the Fig S30, the absorption spectrum of **G1/Py** lacks CT band and the maximum peaks of **G1** (see Fig 2C) and **G1/Py** are nearly identical, indicating an absence of CT in ground state. Since the weak association of excited **G1/Py**, the emission of exciplex tends to be quenched. For host-guest system, CB[8] can act as an ideal confined container to stabilize **G1/Py** in both ground and excited states, thus showing a considerable variation in absorption and emission spectra. Relevant references such as ACS Appl. Mater. Interfaces 2019, 11, 14399–14407 may help to understand this issue further.

Response:

As we know, the characteristics of exciplex emission are: (1) concentration-dependent emission, (2) broad and featureless emission signal, and (3) longer lifetime and longer emission wavelength than the monomer (*Chem. Rev.* **2011**, *111*, 7260; *New J. Chem.* **2018**, *42*, 11249; *Adv. Mater.* **2015**, *27*, 2378). Although the CT interaction shares similar characteristics, it typically generates a new absorption signal in the visible region, and the signal in excitation spectrum of CT emission matches with the CT absorption signal.

After carefully re-evaluating the emission assignment of **Py/G1/CB[8]** in aqueous solution or solid state, we found that: 1. At high concentration (1.0 mM), the aqueous solution of **Py-G1** shows a weak emission at 514 nm with a large full width at half maxima (FWHM) and a lifetime of 37.97 ns (relatively, the excimer emission of **Py** (1.0 mM) is a broad signal at ~488 nm with a lifetime of 86.68 ns) (Supplementary Fig. 35); 2. At low

concentration (0.02 mM), only the monomer emission of **Py** was seen in the mixture of **Py** and **G1**; 3. No obvious charge transfer absorption signal is observed in the UV-vis absorption spectrum of **Py·G1**; 4. The signals in excitation spectrum of **Py·G1** collected by monitoring the emission at 514 nm are in the UV region (Supplementary Fig. 35e); 5. The CT absorption signal could not be seen clearly at lower concentration (Supplementary Fig. 33), but at higher concentration, **CB[8]·Py·G1** shows an apparent CT band at ~ 485 nm (Supplementary Fig. 35c); 6. The signal in excitation spectrum of **CB[8]·Py·G1** collected by monitoring the emission at 560 nm shows a maximum at ~ 490 nm (Supplementary Fig. 35d), which closely matches with the corresponding absorption band of **CB[8]·Py·G1**.

Summary of photophysical data of **Py**, **Py+G1** (1:1), and **CB[8]·Py·G1** in aqueous solution and solid state.

Sample	Concentration / mM	λ_{em} / nm	$\tau_{ave.}$ / ns	λ_{ex} / nm	λ_{em} / nm	$\tau_{ave.}$ / ns
		aq.	aq.	aq.	solid	solid
Py	1.0	488	86.68	-	415 / 499	3.71 / 12.40
G1	0.02	601	0.08	-	566	4.17
Py·G1	1.0	514	37.97	230/304	573	3.21
Py·G1	0.02	391	4.92	-	-	-
CB[8]·Py·G1	0.5	560	0.42	~ 490	569	4.43

From the above data, we believe that:

- As the reviewer suggested, the emission of **Py·G1** at 514 nm in solution could be assigned to an exciplex emission (especially according to 4);
- The emission of **CB[8]·Py·G1** at 560 nm in solution is caused by the HSCT interactions (especially according to 5 and 6);
- In solid state, the slight red-shifted absorption and emission of **Py·G1** compared to individual **Py** and **G1** suggest a charge transfer interaction between **Py** and **G1** in the solid sample **Py·G1** (Supplementary Fig. 38). Additionally, the excitation spectrum of **Py·G1** in solid state shows a maximum signal at ~ 520 nm, which differs significantly from that of individual **Py** and **G1** (Figure R1) and matches with the red-shifted absorption of **Py·G1** in solid state (Supplementary Fig. 38a). The different luminescent behavior between the solution ($\lambda_{em} = 514$ nm, $\tau = 37.97$ ns) and solid state ($\lambda_{em} = 573$ nm, $\tau = 3.21$ ns) also indicates different emission pathways.

Figure R1. Normalized excitation spectra of **Py**, **G1** and **Py-G1** in solid state.

Although all of the data support our conclusion/assumption, we are not certain how CB[8] could switch the exciplex **Py-G1** to CT heterodimer in solution (perhaps confinement from the cavity) and what causes the switch of **Py-G1** from exciplex in solution to CT dimer in the solid state (perhaps distance/packing). According to previous reports, both exciplex and charge transfer pathways are sensitive to the external environment (such as the polarity of solvents) to a certain extent (*J. Am. Chem. Soc.* **2020**, *142*, 7920; *J. Phys. Chem. A* **1999**, *103*, 4993; *New J. Chem.* **2014**, *38*, 2233). We will continue to explore the intrinsic mechanism of the transition from exciplex to charge transfer upon binding.

We have revised this section in the updated manuscript: Due to the strong blue emission of **Py** in diluted solution (2.0×10^{-5} M), the weak emission signal of **G1** is hardly visible in the 1:1 mixture of **Py** and **G1** under the same experimental condition (Fig. 6d). At high concentration (1.0×10^{-3} M), the mixture of **Py** and **G1** shows a weak emission band at 514 nm with a broad full width at half maxima and a long lifetime ($\tau = 37.97$ ns) (Supplementary Fig. 34-35), which could be ascribed to the formation of the exciplex **Py-G1**. A broad emission band at 560 nm emerges and is enhanced by gradually adding CB[8] (from 0 to 1.0 equiv.) to the 1:1 mixture of **Py** and **G1** even in diluted aqueous solution (Fig. 6d). The excitation spectrum of CB[8]·**Py-G1**, collected by monitoring the emission at 560 nm, shows a maximum at about 490 nm, which closely matches with the corresponding charge transfer absorption band of CB[8]·**Py-G1** (Supplementary Fig. 35). Therefore, the newly generated yellow emission at 560 nm could originate from the HSCT

interaction. We believe that the heteroternary complexation narrows the distance and corrects the orientation between **Py** and **G1**, resulting in the stronger charge transfer interaction between **Py** and **G1**.

2. The authors should examine the emission curve of **G1+Py** again because neither weak charge-transfer nor exciplex could reasonably explain this spectrum. Since the emission of **G1+Py** is rather weak, the trace fluorescent impurities may interfere test result. Authors can compare the excited spectrum with the absorption one, if the two spectra don't fit well, the possibility of impurity cannot be ignored.

Response:

We have repeatedly tested the emission of different batches of **G1+Py** (1:1) samples, and all of these samples show the emission at 514 nm. And we have also compared the excited spectrum of **Py+G1** (1:1) with the absorption one, and the two spectra fit well with each other in diluted aqueous solution. After careful evaluation, we have reassigned the emission of **Py·G1** at 514 as exciplex emission (please also check our reply to the comment 1).

3. Authors explained the emission enhancement of **G1** upon addition CB[8] that "is mainly due to the restricted intramolecular rotation of the chromophore (AIE effect) by the confinement effect from the cavity of CB[8]". This explanation is reasonable but probably not sufficient. An additional possibility is that the low polarity of CB[8] cavity could also contribute to the planarization of **G1** and weaken the intramolecular charge-transfer (such as TICT, which can extremely suppress emission) in aqueous condition. Authors can record the QY of **G1** in low polar solvents such as dichloromethane to examine this issue.

Response:

Thanks for the suggestion. We have measured the QY of **G1** in low polar solvent such as dichloromethane and polar solvent such as acetonitrile and dimethyl sulfoxide. The results prove that **G1** shows a higher QY (8.6 %) and shorter emission wavelength (560 nm) in DCM, and much lower QY (0.3 %) with red-shifted emission in acetonitrile, suggesting the twisted intramolecular charge transfer (TICT) nature of **G1**.

Since the host-guest complexation between CB[8] and **G1** leads to the planarization of **G1** (evidenced by crystal structure analysis), we believe the origin of enhanced emission of CB[8]·**G1**₂ is indeed related to the weakened TICT effect. We have added the following two descriptions in the revised manuscript: Compound **G1** exhibits a higher Φ_F value (8.6 %) and shorter emission wavelength (560 nm) in dichloromethane, and a significantly

reduced Φ_F value (0.3 %) with longer emission wavelength (611 nm) in acetonitrile (Supplementary Fig. 11). These observations suggest its twisted intramolecular charge transfer (TICT) nature. Furthermore, the negligible fluorescence of **G1** in diluted aqueous solution with the maximum emission wavelength at 601 nm ($\Phi_F < 0.1\%$) is greatly enhanced upon addition of CB[8] (Fig. 3d), with an Φ_F of 0.71% for CB[8]-**G1**₂. This enhancement can be attributed to the restricted intramolecular rotation of the chromophore (AIE effect) by the confinement effect from the non-polar cavity of CB[8] and the weakened TICT interaction caused by complexation.

Supplementary Figure 11. (a) Photographs of **G1** in different solvents under the excitation of 365 nm UV light (the fluorescence quantum yields of **G1** in different solvents are presented). (b) Normalized emission spectra of **G1** in different solvents ($\lambda_{ex} = 400$ nm).

4. Support theoretical calculation (such as DFT) and improve the single-crystal drawing can help this manuscript to step up a level.

Response:

Thanks for your suggestion. We have carried out the theoretical calculation suggested by Reviewer #1, using the heteroternary complex CB[8]-**Py**-**G1** as a model. The optimized structures obtained by calculation give two different heteroternary complex model (named CB[8]-**Py**-**G1**- α and CB[8]-**Py**-**G1**- β). Notably, CB[8]-**Py**-**G1**- β , which has a planar **G1**, shows a red-shifted emission compared to CB[8]-**Py**-**G1**- α , which has a more twisted **G1**. This is consistent with the observation that grinding causes the red-shifted emission of the

ternary complexes. We have added this discussion in the revised manuscript, and the related data are included in the Supplementary information. Please also refer to our response to comment 4 raised by Referee#3 for detailed information on the theoretical calculations.

The presentation of single crystals in Figure 4-6 have been improved for clarity. The information we want to highlight in crystal structure are (1) the conformation of guests, (2) host-guest binding modes, and (3) the intermolecular interactions, particularly the π -stacking of the guests inside the cavity. Therefore, we have revised the single-crystal drawings by updating the style of the inclusion complex to better illustrate the binding mode. The different conformations of **G1** in CB[8]·**Py**·**G1** are indicated with different colors in Figure 6. We hope that these revised figures could provide a clearer picture of the crystal structures.

5. Compared with **G1** and **G3**, only CB[8]·**G2**₂ could not return to its initial state after fuming with water. Please explain the reason.

Response:

After carefully analyzing the crystal structure, we found that there are significantly more hydrogen bonds, especially C-H...O interactions, in the CB[8]·**G2**₂ crystal (15 types of C-H...O interactions, see Figure R2) compared to CB[8]·**G1**₂ (7 types of C-H...O interactions, see Supplementary Figure 26) and CB[8]·**G3**₂ (10 types of C-H...O interactions). This could be attributed to the two aldehyde groups present in **G2**. We hypothesize that the fuming process leads to a recrystallization of CB[8]·**G2**₂, and the abundance of hydrogen bonds is favorable for the formation of the crystalline state with regular packing rather than its initial amorphous state. In contrast, the amorphous samples of CB[8]·**G1**₂ and CB[8]·**G3**₂ could not undergo such a recrystallization process by fuming with water, owing to the fewer intermolecular interactions, remaining the amorphous state and showing reversible spectral change.

The amorphous solid of CB[8]·**G2** is obtained by lyophilization, while the crystalline solid sample is obtained by fuming the ground sample of CB[8]·**G2** with water. Relative to CB[8]·**G1**₂ and CB[8]·**G3**₂, it appears that the emission of CB[8]·**G2**₂ is more sensitive to the sample preparation method. Fuming with water results in the formation of crystalline CB[8]·**G2**₂ instead of its initial amorphous state caused by lyophilization, thereby leading to the irreversible mechanochromic behavior.

Figure R2. Multiple intermolecular hydrogen bonds in (a) CB[8]-**G22-a** and (b) CB[8]-**G32-a** (the hydrogen bonds between a single inclusion complex and adjacent molecules are presented).

Reviewer #2 (Remarks to the Author):

In the manuscript, the authors described a host-guest inclusion-directed strategy to construct sequentially red-shifted mechanochromic materials. A series of TPA derivatives were generated, and then used for the MCL property comparison with their complexes stabilized with CB[8]. The report is impressive, but still need some improvement before the acceptance:

1. "**G1** is highly emissive in solid state, exhibiting yellow fluorescence at 566 nm with an Φ_F of 32.47%, and the maximum emission wavelength is red-shifted to 578 nm ($\Phi_F = 21.64\%$) after grinding. Powder X-ray diffraction (PXRD) results disclose that the sharp and intense diffraction peaks of **G1** powder is weakened significantly by grinding (Supplementary Fig. 11b), suggesting that grinding could partially destroy the ordered packing arrangement". Actually, red-shifted emission is corresponding to the stronger packing or aggregation. What is the real relationship between the red shifted emission and the amorphous state?

Response:

The red-shifted emission of **G1** after grinding can be attributed to the more planarized conformation, and the enhanced π - π interaction also contribute to this spectral change. As we know that other organic mechanochromic small molecules, especially twisted AIE dyes, their crystalline samples usually show a shorter emission wavelength owing to the organized arrangement, rigid packing and twisted conformation. While the same species

in amorphous state possess a red-shifted emission with disordered arrangement, planar conformation and enhanced π -conjugation degree/ π - π interactions (*J. Am. Chem. Soc.* **2016**, *138*, 12803; *J. Mater. Chem. C* **2018**, *6*, 6327; *J. Mater. Chem. C* **2022**, *10*, 14834).

Grinding of **G1** leads to a morphological change from crystalline to amorphous state, and the amorphous state of **G1** corresponds to a more planarized conformation and enhanced π - π interaction, further resulting in the red-shifted emission.

2. The excitation wavelength values should be listed with the emission data.

Response:

We have added the excitation wavelength values for all emission spectra in both main text and supplementary information.

3. Since Φ_F value detection always adopts error about 10 %, too many significant digits shown in the value are meaningless. 2 significant digits are enough in most cases.

Response:

We agree and have corrected the significant digits of the Φ_F value in our manuscript (2 significant digits are presented in the revised text).

4. About the binding constant calculation shown in Fig.2, what is the expansion of the important signals shown in high resolution mass spectrum? Does it fit well with the simulated data? What are observed in fluorescence and UV-vis titration after the molar ratio of CB[8] larger than 0.5 molar equiv.? How about the binding constants from fluorescence and UV-vis titration? And how about the other thermodynamic parameters from ITC? The comparison of the binding constants from different methods related to same reasonable equilibriums should be listed.

Response:

HRMS We have added the expansion of the important signals shown in high resolution mass spectrum in Figure 3b, and the experimental data fits well with the simulated one. The expansions of the important signals of CB[8]·**G2₂** and CB[8]·**Py·G1** are also provided in the revised supplementary information (Supplementary Fig. 18 and 32).

Figure 3b. ESI-HRMS spectrum of CB[8]·G1₂.

Fluorescence and UV-vis titration We have conducted the fluorescence and UV-vis titration of **G1** with over 0.5 equiv. of CB[8]. Upon adding CB[8], the binding equilibrium ($\text{CB[8]} + 2\text{G1} \leftrightarrow \text{CB[8]}\cdot\text{G1}_2$) gradually shifts towards $\text{CB[8]}\cdot\text{G1}_2$, resulting in spectral changes. Both the absorption and emission spectra undergo slight changes with addition of more than 0.5 equiv. of CB[8]. The trend of change is consistent with that observed with no more than 0.5 equiv. of CB[8]. This is quite normal and also shows that the host-guest binding ratio (1:2) does not change with the amount of CB[8].

Binding constant In our manuscript, we present the binding constant of the complex $\text{CB[8]}\cdot\text{G1}_2$ in order to (1) demonstrate the high stability of the inclusion complex, (2) show the rationality of obtaining solid samples of inclusion complex through lyophilization. Since determining/comparing the k_a value obtained through different methods is not the focus of this work, we will not to discuss or compare the methods for determining k_a any further. The k_a values obtained by ITC experiments are close to those reported for CB[8]-based 1:2 host-guest complexes (*Angew. Chem. Int. Ed.* **2016**, *55*, 15915; *Chem. Sci.* **2017**, *8*, 8357). Therefore, we believe that our proposed k_a is convincing and suitable.

ITC data The thermodynamic parameters from ITC including ΔH , ΔS , and ΔG are provided in the revised supplementary information (Supplementary Table 1). We have added the following discussion in the revised manuscript: **According to the thermodynamic data obtained from ITC, the host-guest complexation is mainly governed by enthalpy.**

5. The text should be improved. For example, line 18, “intermolecular change transfer” should be “intermolecular charge transfer”. Line 19, “longer emission than that of free guests (even longer than ground guests),” should be “longer emission wavelength

compared with free guests in solution, even corresponding pure guests in solid state.” Line 31, “triphenylamine (TPA) and tetraphenylethylene derivatives and others” should be “triphenylamine (TPA), tetraphenylethylene derivatives, and others”. And more other writing parts should be careful checked and improved.

Response:

Many thanks for pointing out those mistakes/typos. The mistakes and other typos in the manuscript as well as in supplementary information have been carefully checked and revised. For Line 19 in the abstract, since we would like to highlight the sequentially red-shifted emission in solid state, this sentence is revised as: “longer emission than that of free guests in the solid state (even longer than ground guests)”.

Reviewer #3 (Remarks to the Author):

In the manuscript entitled “Title: The Host-guest Inclusion Driven by Host-stabilized Charge Transfer for Construction of Sequentially Red-shifted Mechanochromic System” the effect of host-guest interaction in influencing the mechanochromic properties of organic molecules has been explained by the authors. Authors have used a pyridine functionalized with triphenyl amine and have formed homoternary inclusion complexes with cucurbit[8]uril (CB[8]) by host stabilized intermolecular charge-transfer interaction. The molecules **G1-G3**, exhibited red-shifted emission upon grinding in the solid-state which is ascribed to increase in the planarity of conformation induced by grinding. The complexation with CB[8] has further red-shifted emission maxima pertaining to the host stabilized intermolecular charge-transfer interaction. Grinding of this complex has further shifted the emission towards higher wavelength. Authors have done extensive studies to characterize the complexation and further spectroscopic properties to justify the mechanochromic behavior. The manuscript has enough novelty and suitable to publish in this journal. However, the author should address the following points.

1. Authors have observed a difference in the mechanochromic properties of **G3** both in solid state and in complexed state with CB[8]. For example, the PXRD pattern of solid ground **G3** obtained after fuming with water is different from the PXRD pattern of **G3** before grinding. Similar observations were shown by **G3** complexed with CB[8] as well. How does the authors account for this difference in behaviour of **G3** compared to **G1** and **G2**.

Response:

The PXRD pattern of the fumed **G3** sample is different from that of the initial one, indicating that the fumed sample is in a new crystalline state (Supplementary Fig. 14). This new crystalline state corresponds to a different emission wavelength compared to the initial and ground samples. The phenomenon of polymorphism-dependent emission has also been observed in other organic dyes (*Angew. Chem. Int. Ed.* **2018**, *57*, 12473; *Angew. Chem. Int. Ed.* **2020**, *59*, 9972; *Angew. Chem. Int. Ed.* **2020**, *59*, 3739). As for CB[8]-**G3**₂, the PXRD pattern of the fumed sample only shows several diffraction signals with quite low intensity (this subtle change in the PXRD pattern is not remarkable enough to conclude that the fumed sample is in a different morphological state). Moreover, the emission wavelength could be restored after fuming with water.

Actually, previous studies have shown that (1) the mechanochromic behavior of organic molecules, including reversibility, is highly dependent on their conformation and packing arrangement, and (2) even a subtle change in the chemical structure of organic mechanochromic molecules could significantly affect their conformation and/or packing arrangement, leading to a different mechanochromic behavior. Although crystal structures of **G2** and **G3** are not obtained in our study, the differences in their chemical structures are likely responsible for the varied mechanochromic behavior observed, especially in the case of **G3**, which has two cationic sites.

2. “Also, **G1** shows an obvious increased fluorescence lifetime upon binding with CB[8]”. The lifetime decay plot is missing in both main text and SI. Authors should show the decay profile.

Response:

The decay profiles of all the time-correlated single photon counting (TCSPC) experiments have been added in the revised Supplementary information (Supplementary Fig. 15-16).

3. The authors should account the reason for decrement in the absorbance for π - π^* band on forming the ternary complex in all the cases.

Response:

The observed decrease in absorbance (hypochromism) for the ternary complex is attributed to a lower refractive index inside the cavity of CB[n] compared to bulk water such that the oscillator strength for the electronic transition (S_0 - S_1) reduces significantly for the inclusion complex compared to the free guests in bulk water (*J. Phys. Chem. B* **2021**, *125*, 7946; *J. Phys. Chem. B* **2016**, *120*, 11266). This decrement in absorbance for

the π - π^* band upon formation of the host-guest inclusion complex is a common phenomenon observed in almost all CB-based host-guest systems (*Chem. Sci.* **2020**, *11*, 812; *Chem. Commun.* **2020**, *56*, 655; *Adv. Optical Mater.* **2023**, *11*, 2202431; *Adv. Mater.* **2022**, *34*, 2203534). Since the absorbance of π - π^* band is not unique to this system, we have chosen not to include this explanation in the main text.

4. Authors have assumed that since the triphenyl amine is located outside the cavity, the complex is sensitive to mechanical stimuli. Could this statement be made more convincing by elucidating in a quantitative manner or by computational studies?

Response:

Thanks for your valuable suggestion. Theoretical calculations are a powerful tool to help us understand the MCL behavior and make our suppositions more supportive. We have chosen the heteroternary complex CB[8]·Py·G1 as a model among the inclusion complexes for theoretical calculation.

The ground state geometry optimization is conducted using the B3LYP-D3/def2SVP level, and the first excited state geometry optimization is conducted using the wB97X-D3/def2SVP level. The calculation methods are adapted from a related ternary host-guest inclusion system (*ACS Materials Lett.* **2023**, *5*, 1227).

Two optimized structures of the heteroternary complex CB[8]·Py·G1 (named CB[8]·Py·G1- α and CB[8]·Py·G1- β) are built based on the single crystal of CB[8]·Py·G1. Among them, G1 in CB[8]·Py·G1- β is more planar than that in CB[8]·Py·G1- α (Supplementary Fig. 42). Importantly, CB[8]·Py·G1- β with a planar G1 shows a red-shifted emission relative to CB[8]·Py·G1- α with a more twisted G1, suggesting that the planar conformation of G1 could be responsible for the red-shifted emission under mechanical stimuli. This calculation result is in line with the experimental observation of the red-shifted emission of the heteroternary complex CB[8]·Py·G1 upon grinding (and could also be used to explain the red-shifted emission of the homoternary complexes upon grinding). Although the calculated λ_{em} values (399 nm for CB[8]·Py·G1- α and 405 nm for CB[8]·Py·G1- β) are much different from the experimental value, the trend of emission obtained by calculations is in agreement with the experimental results (similar results reported in *ACS Materials Lett.* **2023**, *5*, 1227). The independent gradient model (IGM) analysis for CB[8]·Py·G1- α/β clearly reveals the existence of multiple noncovalent interactions (the green region) between the host and guests (Supplementary Fig. 43), which is in agreement with the X-ray crystallographic analysis. The Cartesian coordinates of the optimized CB[8]·Py·G1- α/β have been added to the revised Supplementary

information (Supplementary Table 6, 7). Considering the detailed single-crystal information and adequate photophysical data of the inclusion complexes, we now believe that our conclusions/suppositions are more convincing with the above theoretical calculations.

Supplementary Figure 42. (a) Optimized structures of CB[8]·Py·G1 (CB[8]·Py·G1- α and CB[8]·Py·G1- β). (b) Conformation of G1 in CB[8]·Py·G1- α (Calculated λ_{em} = 399 nm). (c) Conformation of G1 in CB[8]·Py·G1- β (Calculated λ_{em} = 405 nm).

Supplementary Figure 43. Independent gradient model (IGM) analysis for CB[8]·Py·G1- α (a) and CB[8]·Py·G1- β (b). The green surfaces represent the noncovalent interactions.

We have added the following discussion in the revised manuscript: **Theoretical calculations were then conducted to gain a deeper insight into the MCL behavior of the**

heteroternary complex. As shown in Supplementary Fig. 42, two optimized structures of heteroternary complex, (CB[8]·Py·G1)- α and (CB[8]·Py·G1)- β , were built based on the single crystal of CB[8]·Py·G1. Among them, G1 in (CB[8]·Py·G1)- β is more planar than that in (CB[8]·Py·G1)- α . Importantly, (CB[8]·Py·G1)- β with a planar G1 exhibits a red-shifted emission compared to CB[8]·Py·G1- α with a more twisted G1, suggesting that the planar conformation of G1 may be responsible for the red-shifted emission under mechanical stimuli. This calculation result is in line with the experimental observation of the red-shifted emission of the heteroternary complex CB[8]·Py·G1 upon grinding. The independent gradient model (IGM) analysis for (CB[8]·Py·G1)- α/β clearly reveals the existence of multiple noncovalent interactions (the green region) between the host and guests (Supplementary Fig. 43), which is in agreement with the X-ray crystallographic analysis. This calculation result could also explain the red-shifted emission of the homoternary complexes upon grinding, since the TPA groups of G1-G3 in the homoternary complexes are also located outside the cavity, as shown in the crystal structures.

5. Since there is a charge-transfer interaction between Py and G1, authors should also confirm the same in solid state by comparing the individual absorption and emission spectra of Py and G1 with Py·G1.

Response:

As shown in Supplementary Fig. 38, the solid Py·G1 exhibits a red-shifted and enhanced absorbance in the visible region, and a slight red-shifted emission is also observed (Supplementary Fig. 38), suggesting the presence of charge transfer interaction between Py and G1 in the solid Py·G1. We have added the following description in the revised text: The powder shows slightly red-shifted absorption and emission compared to individual Py and G1 (Supplementary Fig. 38), suggesting the existence of D-A structure.

Supplementary Figure 38. Normalized absorption (a) and fluorescence emission spectra (b) of Py, G1 and Py·G1 in solid state.

6. It is intriguing to see that the mixture **Py:G1** is insensitive to mechanical stimuli, while the same complex inside CB[8] is mechanochromic. How does the authors justify this observation. Also, how is the lifetime of respective emission band changing from 1:1 mixture of **Py:G1** to the 1:1:1 mixture of **Py:G1:CB[8]**.

Response:

We propose that the strong association of between **Py** and **G1** in solid state leads to a rigid packing of **Py·G1**, further preventing the mechanical stimuli from altering the arrangement/conformation, resulting in poor MCL property of **Py·G1**. In contrast, the steric effect of CB[8] in the heteroternary complex CB[8]·**Py·G1** certainly leads to a more isolated D-A structure and a relatively loose arrangement, allowing for greater conformation variation upon grinding and endowing the complex with better MCL performance. The solid state absorbance of CB[8]·**Py·G1** shows an obvious redshift after grinding (Supplementary Fig. 41), indicating the conformation variation process (*Chem. Commun.* **2020**, 56, 13638; *Angew. Chem. Int. Ed.* **2021**, 60, 12443). Theoretical calculations suggest that the optimized structures of the heteroternary complex CB[8]·**Py·G1**- β , with a less twisted **G1**, show a longer emission wavelength than that of CB[8]·**Py·G1**- α with a more twisted **G1** (Supplementary Fig. 42), validating that the MCL behavior of CB[8]·**Py·G1** is highly related to the conformation variation.

Grinding-induced destruction of a large amount of intermolecular interactions, particularly intermolecular hydrogen bonds between adjacent inclusion complexes (Supplementary Fig. 37), may synergistically induce a larger spectral shift compared to the D-A structure without a host (*J. Mater. Chem. C*, **2021**, 9, 17307). Several diffraction peaks of CB[8]·**Py·G1** disappear entirely after grinding, indicating that the partially ordered packing of CB[8]·**Py·G1** is destroyed.

In the solid state, the fluorescence lifetimes of **Py·G1** and CB[8]·**Py·G1** are measured to be 3.21 ns and 4.43 ns, respectively (Supplementary Fig. 16 g-h). This slight increase in lifetime could be mainly due to the enhanced charge transfer interaction of the heteroternary complex. CB[8] acts as an ideal container to stabilize the excited state, which is also responsible for the longer lifetime of the inclusion complex (*Chem. Rev.* **2022**, 122, 9032).

7. Authors may refer to other kinds of host stabilized charge-transfer interaction and refer them in the manuscript. *J. Am. Chem. Soc.* **2022**, 144, 10854.

Response:

After checking, we believe the work mentioned by Referee#3 is highly related to our manuscript. We have cited this work in the revised text as ref. 44.

8. There are certain typos in the main text and supporting information (SI); for instance, in the main text page 10, line 182, 'eimssion' has to be replaced by 'emission', page 11, line 211, 'amphorization' has to be replaced by 'amorphization', page 14, line 276, 'encapsated' to be replaced by 'encapsulated', and, page 17, line 277, 'ledas' to 'leads'. Authors should correct these typos.

Response:

We are so sorry for those mistakes/typos. In the revised text, as well as in supplementary information, we have carefully corrected those issues.

REVIEWERS' COMMENTS

Reviewer #1 (Remarks to the Author):

This revised version can be accepted.

Reviewer #2 (Remarks to the Author):

The authors improved the manuscript fully following the commands from the reviewers. Now it is good enough to be accepted by Nature Communications.

Reviewer #3 (Remarks to the Author):

Authors have addressed all my concerns and the manuscript can be accepted as such.